# Estimating the causal effects of work-related and non-work-related stressors on perceived stress level: A fixed effects approach using population-based panel data

**Finn Breinholt Larsen**[1]*, **Mathias Lasgaard**[1], **Morten Vejs Willert**[2], **Jes Bak Sørensen**[1]

**1** DEFACTUM, Central Denmark Region, Aarhus, Denmark, **2** Department of Occupational Medicine, Danish Ramazzini Centre, Aarhus University Hospital, Aarhus, Denmark

* finn.breinholt@stab.rm.dk

## Abstract

### Objectives

Prolonged or excessive stress can have a negative impact on health and well-being, and stress therefore constitutes a major public health issue. A central question is what are the main sources of stress in contemporary societies? This study examines the effects of work-related and non-work-related stressors and perceived social support on perceived stress within a causal framework.

### Methods

Panel data were drawn from two waves (2013 and 2017) of the population-based health survey "How are you?" conducted in the Central Denmark Region. The analytical sample comprised 9,194 subjects who had responded to both surveys. Work-related and non-work-related stressors included major life events, chronic stressors, daily hassles and lack of social support. Perceived stress was measured with the 10-item Perceived Stress Scale (PSS). Data were analysed using fixed effects regression in a fully balanced design.

### Results

The largest effects on PSS were seen in own disease, work situation and lack of social support. Other stressors affecting the perceived stress level were financial circumstances, relationship with partner, relationship with family and friends, and disease among close relatives. Most variables had a symmetrical effect on PSS.

### Conclusions

The results point to the need for comprehensive policies to promote mental health that span life domains and include both the individual and the group as well as organizational and societal levels. The study indicates that there are multiple potential entry points for stress prevention and stress management. However, it also shows that disease, work situation and

**Data Availability Statement:** The data contain potentially sensitive information and that there is a possibility of deductive disclosure. Therefore data

may not be shared publicly according to the Danish Data Protection Law, section 10 (https://www.datatilsynet.dk/media/7753/danish-data-protection-act.pdf). The de-identified data will be made available upon request to DEFACTUM, Public Health and Health Services Research, Central Denmark Region (hvordanhardudet@rm.dk) with an appropriate, restricted use data agreement in place.

**Funding:** The study was funded by the Velliv Foreningen, grant number 19–0176 (https://www.vellivforeningen.dk/). The grant was received by FBL. The funders had no role in the study design, data collection and analysis, decision to publish or preparation of the manuscript.

**Competing interests:** The authors have declared that no competing interests exist.

social support weigh heavily in the overall picture. This points to the healthcare system and workplace as key institutional venues for action.

## Introduction

The term *stressor* refers to any physical, psychological or social event or condition that can cause strain in individuals and, eventually, harm physical health and mental well-being [1]. Stressors may have an episodically, prolonged or chronic nature and range from daily hassles to life-changing events [2–4]. Stressors are key constructs in stress research as they are potential targets for prevention and health-promoting interventions [5].

Research on stressors has, since its inception, specialized in many directions, thereby deepening our understanding of potential causes of stress [6]. But the question still remains: What are the main sources of stress in contemporary societies? To answer this, it is necessary to study multiple stressors simultaneously as pointed out in previous studies [2,6,7]. In particular, studies are needed that include both work-related and non-work-related stressors [7–12]. Work-related stressors have been intensively researched, but they have predominantly been studied in isolation from non-work-related stressors [8,9]. Similarly, studies of non-work stressors such as disease rarely include work-related stressors [13].

The theoretical underpinning of this study is the psychological Transaction Stress Model (TSM) [14] and the sociological Stress Process Model (SPM) [2]. Together, the two theories support an understanding of stress as a psychological but socially situated and generated phenomenon [7].

TSM emphasizes the cognitive aspect of stress. Psychological stress refers to a relationship between the person and the environment where the demands of a particular situation are perceived as taxing or exceeding the person's resources [14]. The key point is that stress is not a direct product of a stressor but of a process of cognitive appraisal including a primary appraisal in which the person determines whether a given stressor threatens the person's well-being or not, and a secondary appraisal in which the person assesses whether he or she has the necessary resources to cope with the threat [14]. As a consequence, exposure to the same objective stressor can prompt very different stress responses between individuals and in the same individual on different occasions.

In contrast, SPM focuses on social phenomena that cause stress. According to SPM, stress is the result of a social process where stressors are largely allocated to groups and individuals on the basis of social characteristics. Stressors are often connected to social roles and rarely act in isolation but combine in an additive or multiplicative manner to produce stress [15].

Despite their merits as a framework for understanding the stress process, neither TSM nor SPM offer any theoretical guidance on which stressors are the main sources of stress, leaving it to empirical investigations to discover [5]. Achieving a better understanding of how different stressors affect stress is a crucial step in the development of effective preventive measures against physical and mental health problems. In a previous study, we used a comprehensive approach addressing a variety of stressor domains in order to uncover the relative importance of work-related and non-work-related stressors and perceived social support on overall perceived stress [7]. The most important explanatory variables were disease, lack of perceived social support and work situation. The study was based on a cross-sectional population sample which limits causal inference due to uncontrolled confounding. In the present study, we used

longitudinal data to analyse within-person variability, thereby explicitly adopting a causal inference framework [16].

One factor that may influence the strength of the relationship between a domain-specific stressor and perceived stress is domain centrality, or the degree to which a specific domain is considered important to the person's life as a whole [17,18]. There is a close connection between the concept of domain centrality and the appraisal of stressors [18]. In this regard, stressors in a domain that a person finds important are more likely to be appraised as stressful. Domain centrality typically changes throughout the life course and is also affected by societal macro trends. For instance, although work remains an important domain, a decline in work centrality in western countries has been observed in population studies since the 1980s, probably caused by increasing individualism, increased demand for work-life balance and the development of a leisure culture combined with economic prosperity [19]. Significant relationships have also been reported between domain centrality and gender, age, education and occupational status [20].

An important theme in stress research is the connection between stressors across different domains of life, i.e. how stressors combine, how layers of stress accumulate and how sequences of stress start and continue [21]. Two key concepts in this context are stress spillover and stress crossover. The former signifies that stress in one domain leads to or exacerbates stress in other domains (e.g. when job loss causes financial difficulties which, in turn, provokes marital disagreements) [17,22]. Stress crossover arises when stress experienced by one individual affects another individual's stress experience (e.g. when stress experienced in the workplace by a parent leads to stress in his or her child) [23,24]. The accumulated effect of exposure to stressors over a long period of time can cause a form of chronic stress and eventually lead to wear and tear on the body, often referred to as an "allostatic load" [25].

Although studies of domain centrality, stress spillover, and stress crossover do not per se aim to uncover the main sources of stress in modern society, by definition they include stressors from two or more societal contexts and have thereby expanded our understanding of the stress process compared to single-factor studies.

Furthermore, the strength of the relationship between a domain-specific stressor and perceived stress may be influenced by the role personal agency plays in the stress process as pointed out by Peggy A. Thoits [26]. According to Thoits, it is important to distinguish between life events as controllable or uncontrollable, voluntary or involuntary and self-initiated or other-initiated events. From this perspective, stressors triggered by one's own choices (although these will often be constrained by external circumstances) may expose individuals to less stress or make them more resilient to stress. As one example of this phenomenon, a study of Asian immigrants in the United States found that well-planned compared to poorly planned migration lowered acculturative stress, and multiple strong reasons for migration buffered the negative effect of acculturative stress on mental health [27]. As another example, a study in the United Kingdom found that people with poor mental health were more likely to self-select from employment into self-employment, which gave them an—albeit short-termed—boost in mental health due to lower work-related strain [28]. This is despite the fact that working for oneself generally involves high levels of stressors and, as a consequence, has been assumed primarily to attract people in good health [29].

Thoits' extension of the stress process model has an important methodological consequence. If a (greater or lesser) part of the exposure to certain stressors is due to self-selection into or out of particular social conditions, and this is controlled for by using longitudinal data and appropriate statistical methods, it can be assumed that the effect of these stressors on the stress response will be somewhat reduced compared with when data are analysed without taking this into account.

## Aim

In the present study, we aimed to estimate causal effects of a range of work-related and non-work-related stressors and perceived social support on perceived stress using fixed effects (FE) regression in panel data. We wanted to identify significant drivers of change in perceived stress at the individual level in order to gain knowledge that can strengthen preventive efforts for better mental health at population level.

## Methods

### Study design and data collection

Self-reported data were drawn from two waves (2013 and 2017) of the population-based health survey "How are you?" conducted in the Central Denmark Region. In both waves, participants were invited to complete a web-based or postal questionnaire. The 2013 survey involved a representative population sample of 54,300 citizens aged 16 years and above, drawn from the Danish Civil Registration System [30]; the response rate was 61%. In order to obtain panel data, a random sample of respondents from the 2013 survey were re-invited for the 2017 survey (n = 10,543). They formed the panel (study sample) together with persons who by chance had been sampled from the population register in both 2013 and 2017 as participants in the cross-sectional health surveys, which are conducted every four years (in total, n = 12,583). The panel response rate in 2017 was 81% (n = 10,203).

Data from the survey were linked with national administrative registers using the unique personal identification number assigned to all Danish citizens [31]. Register data included age, sex and ethnic background.

### Ethics

The study was approved by the Danish Data Protection Agency (r. no. 2012-58-0006) and registered in the internal directory of research projects in the Central Denmark Region (r. no. 1-16-02-352-19). According to Danish law, no formal ethical approval of survey and register-based studies is required from an ethics committee or other research body (§ 14 section 2) [32]. Each participant received written information about the purpose of the survey, and informed consent was obtained from all subjects. All methods were carried out in accordance with relevant guidelines and regulations along with the approval.

### Analytical sample

We define our analytical sample as subjects in the study sample who had responded to both surveys and who had complete data after handling of item non-response (see data analysis) on all exposure (perceived stressors/social support) and outcome measures (perceived stress) (n = 18,388 observations and 9,194 individuals). The data used in FE regression were thus fully balanced. Baseline characteristics of subjects included in the analytical sample are summarized in Table 1. Compared with subjects from the study sample, those not included were on average older with a higher proportion of males, persons with other countries of origin than Denmark, low educational attainment, non-employed, living without spouse/cohabitant, not living with children aged 0–15 years, and with limiting long-term conditions. They also had a higher mean score on PSS. All differences were statistically significant (p<0.05).

### Variables

**Perceived stress scale.** The level of perceived stress was assessed by the 10-item Perceived Stress Scale (PSS) [33]. The PSS is a global measure of stress based on Lazarus' stress model,

**Table 1. Characteristics af the analytical sample at baseline (2013) with Perceived Stress Scale (PSS) score 2013 and 2017.**

| | | | PSS mean (SD) | | PSS within subject variation |
|---|---|---|---|---|---|
| | % | N | 2013 | 2017 | Mean absolute change (SD) |
| **Mean age (SD), years** | 45.5 (17.5) | 9194 | 11.1 (7.1) | 11.5 (6.9) | 4.9 (4.2) |
| **Age, years** | | | | | |
| 16–24 | 14.4 | 797 | 12.1 (5.5) | 12.7 (5.4) | 5.4 (3.6) |
| 25–44 | 33.6 | 2291 | 11.6 (6.3) | 12.0 (6.1) | 5.1 (3.7) |
| 45–64 | 36.0 | 4050 | 10.6 (7.8) | 10.8 (7.5) | 4.6 (4.5) |
| ≥65 | 15.9 | 2056 | 10.0 (7.7) | 11.0 (7.7) | 4.5 (4.7) |
| **Sex** | | | | | |
| Female | 50.2 | 4920 | 11.9 (7.6) | 12.2 (7.5) | 5.1 (4.6) |
| Male | 49.8 | 4274 | 10.2 (6.5) | 10.9 (6.3) | 4.6 (3.8) |
| **Country of origin** | | | | | |
| Denmark | 94.1 | 8912 | 10.9 (7.1) | 11.4 (7.0) | 4.8 (4.3) |
| Other | 5.9 | 282 | 13.9 (5.9) | 14.3 (5.3) | 5.8 (3.4) |
| **Educational attainment** | | | | | |
| Low (0–10 years) | 15.4 | 1410 | 12.3 (7.6) | 12.8 (6.9) | 5.0 (4.3) |
| Medium (11–15 years) | 52.1 | 4943 | 11.0 (7.1) | 11.5 (7.0) | 4.9 (4.3) |
| High (15- years) | 31.4 | 2770 | 10.4 (6.7) | 10.7 (6.6) | 4.7 (4.0) |
| Missing | 1.1 | 71 | 15.1 (6.3) | 17.3 (7.0) | 5.6 (4.6) |
| **Employment status** | | | | | |
| Employed | 62.8 | 5682 | 10.3 (6.5) | 10.8 (6.5) | 4.8 (4.1) |
| Non-employed | 35.2 | 3303 | 12.3 (7.9) | 12.8 (7.5) | 5.0 (4.4) |
| Missing | 2.0 | 209 | 12.9 (8.1) | 13.2 (7.2) | 4.9 (4.4) |
| **Living with spouse/cohabitant** | | | | | |
| No | 23.5 | 2163 | 12.0 (6.2) | 12.4 (6.1) | 5.1 (3.9) |
| Yes | 76.3 | 7012 | 10.6 (7.4) | 11.1 (7.2) | 4.7 (4.3) |
| Missing | 0.2 | 19 | 11.2 (7.1) | 12.6 (6.6) | 5.0 (3.4) |
| **Living with child(ren) aged 0–15 years** | | | | | |
| No | 79.1 | 7446 | 10.9 (7.2) | 11.5 (7.0) | 4.9 (4.3) |
| Yes | 20.7 | 1729 | 11.6 (6.7) | 11.8 (6.8) | 4.8 (3.9) |
| Missing | 0.2 | 19 | 11.2 (7.1) | 12.6 (6.6) | 5.0 (3.4) |
| **Limiting long-term condition(s)** | | | | | |
| Yes | 33.6 | 3176 | 13.1 (7.8) | 13.1 (7.4) | 5.0 (4.2) |
| No | 63.5 | 5720 | 9.9 (6.4) | 10.7 (6.5) | 4.8 (4.2) |
| Missing | 2.9 | 298 | 11.7 (7.5) | 12.6 (7.1) | 4.9 (4.8) |

Note: All percentages, means and standard deviations are weighted. SD: Standard deviation.

designed "to tap the degree to which respondents found their lives unpredictable, uncontrollable and overloading". An increasing sum score (range 0 to 40) indicates increasing perceived stress levels. In the present study, PSS was analysed as a continuous scale.

**Perceived stressors.** Work-related and non-work-related perceived stressors were assessed using nine questions covering daily hassles, chronic stressors and major life events in various life domains, and a single question about perceived social support. The stressor items have been described previously [7]. Both the stressor items and the social support item have four response categories ("no"; "yes, a little"; "yes, partly"; "yes, a lot" respectively "yes, always"; "yes, mostly"; "yes, sometimes"; "no, never or almost never"), which were all used in the analyses.

**Sociodemographic variables.** Age and gender were determined using a combination of self-reported and register data. Country of origin was defined (Denmark/other) using the Danish Civil Registration System [34]. Educational attainment was self-reported and categorised as low (primary school, no further education), medium (upper secondary education, vocational education and/or short higher education) or high (Bachelor's degree or higher level of education) according to the Danish version of the International Standard Classification of Education [35]. Students were categorised according to their expected graduation level. Employment status was assessed (employed/non-employed) using self-reported data.

## Data analysis

We used FE regression models for causal interpretation of the panel data in order to provide knowledge about which stressors contributed most to the observed change in stress within individuals from 2013 to 2017 [36]. The FE regression models remove all observed and unobserved time-invariant confounding in addition to observed time-varying confounding in an effort to obtain a more robust estimate of causal relationships between the independent variables and the outcome than can be achieved by non-causal research designs (e.g. pooled ordinary least squares regression). The causal interpretation of the FE estimates is based on the following assumptions, as summarized by Terrence D. Hill and colleagues: "(1) no unobserved time-varying confounders (classic unobserved heterogeneity), (2) past outcomes do not directly affect the explanatory variables (reverse causality) and (3) past explanatory variables do not directly affect current outcomes (lagged treatments)" [37]. In order to be able to make accurate estimates, the explanatory variables must exhibit sufficient variability because FE estimates are based on changes over time in the individual respondents (within-person variation), and the explanatory variables must be measured reliably [38]. We used PSS as the dependent variable, and stressors and social support as explanatory variables. We compared the estimates of the FE regression model with the estimates of a pooled ordinary least squares (POLS) model, i.e. a model where, unlike the FE model, unobserved heterogeneity is not controlled for.

In order to aid the interpretation of the regression models, the relative importance of the independent variables was determined using dominance analysis [39]. Dominance analysis is a method that decomposes and compares the contribution that each independent variable makes to the explained variance in a regression model. Dominance analysis was applied to the explained within variance ($R^2_{within}$) of the FE regression model, that is, the amount of intra-individual variance of PSS that can be explained by changes in perceived stressors and perceived social support.

Prior to the analyses, data were screened for item non-response. Overall, the analytical sample was reduced by 1,009 respondents from 10,203 to 9,194 due to item non-response (a reduction of 9.9%). Among the respondents in the study sample, PSS item non-response ranged from 2.6% (item 10 in 2013) to 3.9% (item 8 in 2017). If one, two or three items of the PSS scale were missing, the mean of the available items was used to calculate the scale score [40]. If responses to more than three items were missing, the PSS score was regarded as missing which led to exclusion of 566 respondents (2.8%). For stressors, missing values were treated as "no" if respondents had answered at least one of the nine stressor questions. Following this, 993 respondents (8.1%) had not answered the stressor questions and were excluded. Social support was covered with a single question with 202 missing in 2013 (1.6%) and 478 in 2017 (4.7%).

In order to reduce sampling and non-response bias in the 2013 survey, we applied weights constructed by Statistics Denmark using a model-based calibration approach and including sociodemographic characteristics, income, social benefits and healthcare utilisation [41].

Despite a high panel response rate in 2017, panel attrition caused some bias, in the composition of the analytical sample as described above. This was corrected by means of propensity score weighting (PSW) [42]. Using a logistic regression model, we estimated the probability of response in 2017 for each participant. The inverse response probability was used to correct the original weights, multiplying the two values (for technical details see [42–44]). As outcome in the logistic regression model, we used a variable that indicated whether a person in the study sample responded in 2017 or not. Eleven explanatory variables were selected from a gross list of 20 variables chosen on the basis of literature on methods and manual screening. The variables fell into three categories: socioeconomic variables, health variables and variables related to the data collection process. Applying this procedure to the data reduced the differences in socioeconomic composition as well as stress level between the study sample and the analytical sample.

Stata/SE v17.0 (StataCorp, College Station, TX) was used to prepare the data and perform the descriptive and inferential analyses.

## Results

### Descriptive analysis

The mean perceived stress level in the population was 11.1 in 2013 (standard deviation (SD) 7.1) and 11.5 in 2017 (SD 6.9) (Table 1). Although statistically significant (p<0.0003), the practical significance of this increase is negligible judged from Cohen's d effect size, where this corresponds to d = 0.07 for the pooled sample, with d = 0.20 considered the lower threshold for a small effect.

In general, lower levels of stress were found among middle-aged and older respondents, men, respondents with Denmark as their country of origin, respondents with a high level of education, respondents who were employed, respondents who were married/cohabitating and respondents without children in the household or without long-term limiting illness.

Looking at the variation in levels of stress within individuals, the mean absolute change in PSS from 2013 to 2017 was 4.9 (SD 4.2). This corresponds to a value of d = 0.70 for the pooled sample, which amounts to a medium effect size (0.50 ≤ d <0.80). Half of the population had experienced an average increase in PSS of 5.3, while 42% had experienced an average decrease of -5.2; and in 7% the level was unchanged. The correlation between PSS in 2013 and 2017 was moderate (r = 0.58).

Financial circumstances and work situation were the most common stressors in 2013 (reported by 45.4% and 41.6%, respectively), while having been burdened by disease and work situation were the most common stressors in 2017 (reported by 41.9% and 41.6%, respectively) (Table 2). For most stressors and perceived social support, the level of exposure in the population did not change much from 2013 to 2017; the largest change was in financial circumstances, where the proportion not burdened by their finances increased from 54.6% in 2013 to 63.0% in 2017.

Table 2 indicates that there is ample intrapersonal variability in the analytical sample to estimate the effect of the variables of interest with FE regression. Only 580 respondents experienced no change in exposure from 2013 to 2017. On average, changes occurred in 3.4 of the 10 exposure variables.

### Fixed effects model

In the FE model, all perceived stressors are statistically significant predictors of the outcome except for housing and death among close relatives (Table 3). Lack of social support is also significant. Considerable variation is seen in the effect of stressors on PSS. The greatest effects are

**Table 2. Prevalence of stressors and perceived social support and within-person change in exposure, 2013 to 2017.**

| | | N (2013) | % (2013) | N (2017) | % (2017) | Changed | Not changed | % changed |
|---|---|---|---|---|---|---|---|---|
| **Financial circumstances** | | | | | | | | |
| No | | 5627 | 54.6 | 6367 | 63.0 | 713 | 4914 | 12.7 |
| Yes, a little | | 2335 | 27.9 | 1915 | 23.7 | 1450 | 885 | 62.1 |
| Yes, partly | | 738 | 9.9 | 542 | 7.6 | 589 | 149 | 79.8 |
| Yes, a lot | | 494 | 7.6 | 370 | 5.7 | 361 | 133 | 73.1 |
| Total | | 9194 | 100 | 9194 | 100 | 3113 | 6081 | 33.9 |
| **Housing conditions** | | | | | | | | |
| No | | 7715 | 78.6 | 7677 | 78.5 | 869 | 6846 | 11.3 |
| Yes, a little | | 962 | 13.2 | 1001 | 13.6 | 711 | 251 | 73.9 |
| Yes, partly | | 344 | 5.2 | 337 | 5.1 | 300 | 44 | 87.2 |
| Yes, a lot | | 173 | 3.0 | 179 | 2.8 | 139 | 34 | 80.3 |
| Total | | 9194 | 100 | 9194 | 100 | 2019 | 7175 | 22.0 |
| **Work situation** | | | | | | | | |
| No | | 5753 | 58.4 | 5929 | 58.4 | 1233 | 4520 | 21.4 |
| Yes, a little | | 2077 | 24.0 | 1978 | 24.7 | 1332 | 745 | 64.1 |
| Yes, partly | | 800 | 9.9 | 786 | 10.3 | 648 | 152 | 81.0 |
| Yes, a lot | | 564 | 7.8 | 501 | 6.7 | 433 | 131 | 76.8 |
| Total | | 9194 | 100 | 9194 | 100 | 3646 | 5548 | 39.7 |
| **Relationship with partner** | | | | | | | | |
| No | | 7085 | 75.5 | 6937 | 73.3 | 1097 | 5988 | 15.5 |
| Yes, a little | | 1556 | 17.4 | 1706 | 19.5 | 892 | 664 | 57.3 |
| Yes, partly | | 334 | 4.1 | 344 | 4.5 | 275 | 59 | 82.3 |
| Yes, a lot | | 219 | 3.0 | 207 | 2.8 | 182 | 37 | 83.1 |
| Total | | 9194 | 100 | 9194 | 100 | 2446 | 6748 | 26.6 |
| **Relationship with family and friends** | | | | | | | | |
| No | | 6990 | 73.4 | 7121 | 75.3 | 991 | 5999 | 14.2 |
| Yes, a little | | 1834 | 21.3 | 1717 | 19.9 | 1135 | 699 | 61.9 |
| Yes, partly | | 281 | 4.0 | 265 | 3.5 | 236 | 45 | 84.0 |
| Yes, a lot | | 89 | 1.4 | 91 | 1.3 | 73 | 16 | 82.0 |
| Total | | 9194 | 100 | 9194 | 100 | 2435 | 6759 | 26.5 |
| **Disease** | | | | | | | | |
| No | | 5640 | 61.7 | 5325 | 58.1 | 1515 | 4125 | 26.9 |
| Yes, a little | | 2271 | 23.4 | 2414 | 25.3 | 1345 | 926 | 59.2 |
| Yes, partly | | 772 | 8.4 | 918 | 9.8 | 551 | 221 | 71.4 |
| Yes, a lot | | 511 | 6.5 | 537 | 6.9 | 342 | 169 | 66.9 |
| Total | | 9194 | 100 | 9194 | 100 | 3753 | 5441 | 40.8 |
| **Disease among close relatives** | | | | | | | | |
| No | | 5281 | 59.2 | 5298 | 59.6 | 1652 | 3629 | 31.3 |
| Yes, a little | | 2612 | 26.7 | 2553 | 26.2 | 1633 | 979 | 62.5 |
| Yes, partly | | 897 | 9.6 | 899 | 9.3 | 727 | 170 | 81.0 |
| Yes, a lot | | 404 | 4.5 | 444 | 5.0 | 329 | 75 | 81.4 |
| Total | | 9194 | 100 | 9194 | 100 | 4341 | 4853 | 47.2 |
| **Deaths among close relatives** | | | | | | | | |
| No | | 7480 | 81.8 | 7594 | 83.3 | 1177 | 6303 | 15.7 |
| Yes, a little | | 971 | 9.9 | 905 | 9.3 | 823 | 148 | 84.8 |
| Yes, partly | | 403 | 4.4 | 354 | 3.5 | 377 | 26 | 93.5 |
| Yes, a lot | | 340 | 3.9 | 341 | 3.8 | 311 | 29 | 91.5 |

(*Continued*)

**Table 2.** (Continued)

|  |  | N (2013) | % (2013) | N (2017) | % (2017) | Changed | Not changed | % changed |
|---|---|---|---|---|---|---|---|---|
| Total |  | 9194 | 100 | 9194 | 100 | 2688 | 6506 | 29.2 |
| **Other types of distress** |  |  |  |  |  |  |  |  |
| No |  | 8521 | 92.3 | 8110 | 87.2 | 873 | 7648 | 10.2 |
| Yes, a little |  | 308 | 3.2 | 568 | 6.6 | 247 | 61 | 80.2 |
| Yes, partly |  | 205 | 2.4 | 264 | 3.1 | 188 | 17 | 91.7 |
| Yes, a lot |  | 160 | 2.1 | 252 | 3.1 | 135 | 25 | 84.4 |
| Total |  | 9194 | 100 | 9194 | 100 | 1443 | 7751 | 15.7 |
| **Perceived social support** |  |  |  |  |  |  |  |  |
| Yes, always |  | 5287 | 57.3 | 5682 | 59.9 | 1193 | 4094 | 22.6 |
| Yes, mostly |  | 2863 | 30.1 | 2356 | 25.8 | 1754 | 1109 | 61.3 |
| Yes, sometimes |  | 762 | 9.3 | 830 | 10.2 | 494 | 268 | 64.8 |
| No, never or almost never |  | 282 | 3.4 | 326 | 4.1 | 208 | 74 | 73.8 |
| Total |  | 9194 | 100 | 9194 | 100 | 3649 | 5545 | 39.7 |

Note: All percentages are weighted.

Considerable within-person change in stressor exposure and perceived social support was observed from 2013 to 2017. The largest change in exposure was observed in relation to being burdened with disease among close relatives and being burdened by own disease, where 47.2% and 40.8% of the population, respectively, changed exposure status between the four exposure categories. The pairwise strength of correlation between 2013 and 2017 for each type of exposure ranged from moderate (r = 0.52) to very weak (r = 0.09).

seen in own disease and work situation, where PSS increased by 4.8 and 2.9 points, respectively, in people who changed their exposure status from "not" being burdened to being burdened "a lot" from 2013 to 2017. The model explains 19% of the within-variation. Disease, work situation and social support contributed more than 60% of the explained within-variation with 33%, 18% and 11%, respectively.

Comparison of the FE model with the POLS model provides insight into what happens when checking for unobserved heterogeneity. In the POLS model, all stressors and social support are significant, and the regression coefficients are greater; compared with POLS, all coefficients in the FE model are sizeably smaller. In addition, the relative contribution to the explained variance is greater for disease and work situation in the FE model than in the POLS model, while the relative contribution from a number of other variables—most notably social support—is reduced.

It is important to be aware that the effect of changes in a stressor can be asymmetric, e.g. the onset of a stressor can cause an increase in the stress level of a person that is greater than the reduction that occurs when the stressor disappears. To examine this, we tested for asymmetric causal effects in FE regression using a method recently developed by Paul D. Allison based on a solution proposed by York and Light [45,46]. This analysis showed that the magnitude of the effects of onset and end of exposure did not differ significantly in eight out of ten cases. One exception was 'relationship with partner', where going from being very burdened to not being burdened reduced PSS by 4.2 points (p < .000), whereas going in the opposite direction only increased PSS by 0.6 points (p = 0.688) (adjusted Wald test $F_{1, 9193} = 4.95$, p = 0.0261). The second exception was 'other types of distress', where going from partially burdened to not being burdened reduced PSS by 1.8 points (p = 0.002), whereas moving in the opposite direction also reduced PSS but only by 0.2 points (p = 0.743) (adjusted Wald test $F_{1, 9193} = 6.29$, p = 0.0261).

**Table 3. Impact of stressors and perceived social support on perceived stress—comparison of pooled OLS and fixed effects regression models.**

| | Pooled OLS regression | | | Fixed effects regression | | | Change in the size of coefficients POLS -> FE (%) | SE ratio FE/POLS | Relative contribution to explained variance (%) | |
|---|---|---|---|---|---|---|---|---|---|---|
| | Coef. | P | SE | Coef. | P | SE | | | POLS | FE |
| **Financial circumstances** | | *** | | | *** | | | | 9.2 | 7.0 |
| Yes, a little | 0.56 | *** | 0.13 | 0.22 | | 0.16 | -60.8 | 1.25 | | |
| Yes, partly | 1.37 | *** | 0.23 | 0.86 | ** | 0.28 | -37.3 | 1.22 | | |
| Yes, a lot | 2.61 | *** | 0.30 | 1.73 | *** | 0.40 | -33.9 | 1.32 | | |
| **Housing conditions** | | *** | | | ns | | | | 5.3 | 2.8 |
| Yes, a little | 0.40 | * | 0.17 | -0.02 | | 0.21 | -104.3 | 1.20 | | |
| Yes, partly | 0.54 | | 0.30 | 0.33 | | 0.32 | -38.4 | 1.07 | | |
| Yes, a lot | 1.57 | *** | 0.40 | 0.69 | | 0.49 | -56.2 | 1.21 | | |
| **Work situation** | | *** | | | *** | | | | 12.3 | 18.5 |
| Yes, a little | 1.10 | *** | 0.13 | 0.70 | *** | 0.16 | -36.5 | 1.21 | | |
| Yes, partly | 2.70 | *** | 0.20 | 1.77 | *** | 0.24 | -34.4 | 1.21 | | |
| Yes, a lot | 3.54 | *** | 0.28 | 2.79 | *** | 0.33 | -21.3 | 1.17 | | |
| **Relationship with partner** | | *** | | | *** | | | | 6.5 | 8.2 |
| Yes, a little | 0.89 | *** | 0.14 | 0.63 | *** | 0.18 | -29.8 | 1.26 | | |
| Yes, partly | 1.76 | *** | 0.31 | 1.36 | *** | 0.34 | -22.5 | 1.08 | | |
| Yes, a lot | 2.65 | *** | 0.42 | 2.26 | *** | 0.55 | -14.7 | 1.32 | | |
| **Relationship with family and friends** | | *** | | | *** | | | | 10.8 | 8.3 |
| Yes, a little | 1.43 | *** | 0.14 | 0.81 | *** | 0.18 | -43.5 | 1.26 | | |
| Yes, partly | 2.76 | *** | 0.32 | 1.70 | *** | 0.40 | -38.4 | 1.24 | | |
| Yes, a lot | 2.98 | *** | 0.66 | 1.93 | ** | 0.73 | -35.3 | 1.10 | | |
| **Disease** | | *** | | | *** | | | | 27.5 | 32.0 |
| Yes, a little | 1.61 | *** | 0.12 | 0.97 | *** | 0.16 | -39.8 | 1.31 | | |
| Yes, partly | 4.28 | *** | 0.21 | 2.89 | *** | 0.27 | -32.5 | 1.26 | | |
| Yes, a lot | 6.11 | *** | 0.29 | 4.87 | *** | 0.36 | -20.3 | 1.26 | | |
| **Disease among close relatives** | | *** | | | * | | | | 4.0 | 2.4 |
| Yes, a little | 0.34 | ** | 0.12 | 0.22 | | 0.14 | -35.4 | 1.20 | | |
| Yes, partly | 0.57 | ** | 0.19 | 0.49 | * | 0.21 | -12.9 | 1.15 | | |
| Yes, a lot | 1.98 | *** | 0.26 | 0.81 | * | 0.33 | -59.1 | 1.28 | | |
| **Deaths among close relatives** | | *** | | | ns | | | | 1.0 | 0.6 |
| Yes, a little | 0.29 | | 0.16 | 0.20 | | 0.18 | -28.8 | 1.14 | | |
| Yes, partly | 0.81 | ** | 0.27 | 0.22 | | 0.28 | -72.5 | 1.03 | | |
| Yes, a lot | 0.22 | | 0.29 | 0.35 | | 0.37 | 59.8 | 1.28 | | |
| **Other types of distress** | | *** | | | *** | | | | 6.8 | 7.4 |
| Yes, a little | 1.46 | *** | 0.23 | 0.83 | ** | 0.30 | -43.4 | 1.27 | | |
| Yes, partly | 1.97 | *** | 0.33 | 0.68 | | 0.37 | -65.4 | 1.12 | | |
| Yes, a lot | 3.68 | *** | 0.41 | 2.61 | *** | 0.43 | -29.1 | 1.04 | | |
| **Social support** | | *** | | | *** | | | | 16.7 | 11.0 |
| Yes, mostly | 1.93 | *** | 0.12 | 1.10 | *** | 0.16 | -43.1 | 1.29 | | |
| Yes, sometimes | 3.80 | *** | 0.20 | 1.94 | *** | 0.26 | -49.0 | 1.28 | | |

(*Continued*)

**Table 3.** (Continued)

| | Pooled OLS regression | | | Fixed effects regression | | | Change in the size of coefficients POLS -> FE (%) | SE ratio FE/POLS | Relative contribution to explained variance (%) | |
|---|---|---|---|---|---|---|---|---|---|---|
| | Coef. | P | SE | Coef. | P | SE | | | POLS | FE |
| No, never or almost never | 3.68 | *** | 0.33 | 1.97 | *** | 0.36 | -46.4 | 1.11 | | |

Note: All figures are weighted. POLS: Pooled ordinary least squares; FE: Fixed effects regression; SE: standard error. All coefficients are tested to be equal to 0 using two-tailed t-tests. Joint tests that all coefficients are equal to 0 for each independent variable are reported on the same line as the variable name

*** p < 0.001

** p < 0.01

* p< 0.05. The relative importance of the independent variables was determined using dominance analysis.

## Discussion

### Main findings

We used FE regression models on panel data to estimate the effects of changes in a wide range of perceived stressors and perceived social support on perceived stress. Changes were in the expected direction, and the effects were symmetrical but of varying strength. Changes in own disease and work situation together with social support had the greatest effects on changes in perceived stress. However, apart from housing and deaths among close relatives, changes in all of the included factors were statistically significant and thus contributed to the changes in perceived stress level. This provides support–as do previous studies–for a comprehensive approach to stress involving a broad spectrum of stressors [7,17,47].

Our findings are in line with longitudinal studies reporting that non-work factors affect workers' mental health [8], work-related psychosocial risk factors are associated with stress-related mental disorders [48] and work-related and non-work stressors affect work performance [49]. However, we have not been able to identify other studies of recent origin with as broad a scope as the present study. Most of the research studying the combined effect of work and non-work stressors focuses on workplace populations or specific professions [50]. In contrast, the present study is based on representative data from a general population with a wide age range including both employed and non-employed respondents. In light of the aforementioned findings, we believe that the present study contributes to better understanding of the main sources of stress in contemporary living. We also believe that our study identifies stressors that could be particularly useful targets for prevention and health promotion interventions aiming at reducing perceived stress both at the individual and population level.

The results of the present study are consistent with our previous cross-sectional study and thus confirm the robustness of the findings when a time dimension is added to the model [7]. However, in the present study, the effects are slightly smaller, which may be due to better confounder control. In the present study, we used a causal design (FE regression), which increases confidence in the results due to removal of confounding from unmeasured characteristics such as genetics, childhood conditions and stable personality traits including propensities to self-selection into or out of particular social conditions [36]. The generalizability of our findings across different contemporary societies will depend on the degree to which our results are mirrored in future studies adopting an approach similar to ours conducted in different countries and with a wide-ranging selection of stressors.

The present study indicates that there are multiple potential entry points for stress prevention and stress management. However, it also shows that disease, work situation and social

support weigh heavily in the overall picture. This points to the healthcare system and the workplace as key institutional venues for action.

The work situation is a well-known stressor [48,50], and the workplace has long been a major arena for stress prevention and management. The present study supports that this should continue to be the case since a poor psychosocial work environment may elicit stress. At the individual and organizational level, improvements in work environment including a better experience of sense of coherence [51], health-promoting management [52] or supervisor support and schedule control [53] may reduce stress associated with work situations. However, the present study as well as previous studies indicate that workplace interventions aimed at reducing psychological stress should address both work and non-work stressors [8,9,54,55]. It is important to discuss how management and society can best act to support employees who feel stressed because of their general living conditions such as a compromised health status due to chronic disease, single parenthood or care tasks for family members [55]. This appears to be neglected even in recent comprehensive research agendas and policy recommendations to promote mental health and well-being in the workplace [56,57].

Like work, chronic illness has long been a known source of stress in stress research which has given rise to concepts such as diabetes-related, cancer-related and skin disease-related stress [58–60]. The pathway from disease to stress starts with an appraisal of the situation as uncertain and a feeling of loss of control, and may end up with stress if coping mechanisms are inadequate [13]. Specific elements of this process may include pain, compromised physical or mental functioning, concerns and burdens related to diagnosis and treatment, doubts about the benefits of treatment in relation to its potential harms and disruption of social roles and relationships [13]. Thus, at the individual level, disease as a stressor may be countered with a changed appraisal of and improved coping with disease-related stressors [61,62]. Social support may also alleviate disease-induced stress [63]. However, stress management is rarely integrated into standard health care [64]. New models for chronic disease management have recently been proposed, e.g. among cancer survivors, with annual mental health and well-being screening, referral if necessary to treatment with cognitive behavioral therapy, and mindfulness-based stress reduction [65–67]. Thoughts along these lines should be combined with more general models of patient-centered health care and health promotion to meet the needs of people with one or more chronic conditions [68].

Social relationships may help us cope with stress, thereby buffering the health effects of stress [69,70]. An understanding of the biological and psychological pathways through which social support might protect against stress is emerging [70,71]. The biological pathways are linked to the autonomic nervous system, the hypothalamic-pituitary-adrenocortical axis and the immune system. Ditzen and Heinrichs refer to experiences of social support as safety signals that affect the biological response to a stressful situation [71]. In the terminology of Lazarus and Folkman, this corresponds to appraisal of a stressor as harmless [14]. Social support both at work and in private life reduces the risk of poor mental health in general [48,72–75], and interventions in the workplace have proven successful [51–53].

It is important to emphasize that other types of stressors–both stressors included in this study and stressors that are not included–are also important, as they can constitute a significant burden in an individual's life or contribute to the total burden and perhaps even be the last straw in a series of events that make it difficult or impossible to deal with the challenges of everyday life [2]. Nevertheless, we find the results important, considering current societal developments.

## Stressors as a part of major societal trends

The significance of our findings should be seen in light of three major societal trends regarding labour force participation, prevalence of chronic diseases and single-person households.

*First*, a trend is seen of growing labour force participation across much of the developed world [76]. For decades, this trend has been driven by younger women, but the ageing of populations has created the need for older persons to remain in the labour force given the expected future labour supply shortage. In many countries, this trend is backed up politically by increases in the statutory retirement age and financial incentives that encourage older people to stay in the labour market. Also, many labour market initiatives have been launched to improve work participation among individuals with chronic conditions–for economic reasons to ensure a sustainable welfare state and for individual reasons because employment is considered an important part of quality of life [76]. As a consequence, more people are exposed to the "hustle and bustle" of working life. Consequently, a growing number of people with different challenges or stressors may be exposed to the work situation as a stressor in daily life. However, a recent review finds that work place interventions may be useful in reducing stress in older workers [77].

*Second*, a trend towards an increase in the prevalence of chronic diseases and multimorbidity is seen [78,79]. Although the ageing of the population explains part of this trend, the prevalence of diseases among younger generations is also increasing [79–82]. In absolute terms, the burden of disease due to chronic conditions is greatest among people under 60 years of age [83]. This means that more people have to deal with illness at the same time as they manage a job. They may be restricted in their ability to work and in their broader daily living. Moreover, managing one or more chronic diseases constitutes an additional workload that can tip the balance between demands on and the capacity of the individual [84]. This may further increase the prevalence and severity of disease as a stressor.

*Third*, a striking growth has been seen in single-person households in many countries during the twentieth century [85]. Single life can have a number of consequences, both positive and negative for the individual [86]. On the negative side, studies show that people who live alone experience, on average, poorer life satisfaction and subjective well-being than cohabiting adults and have lower perceptions of the availability of social support [87]. This societal trend–along with other trends causing increases in social isolation and loneliness–may make more people prone to lack of social support and thus increase the perceived stress level.

The trends mentioned raise concerns that stress levels may increase further, with the mental health of the population deteriorating unless we develop effective strategies to counter the development.

## Strengths and limitations

Our data are well-suited for analyzing changes over time due to the large intrapersonal variability in exposure variables. Despite that, we would like to draw attention to a number of limitations in our study.

The causal interpretation of our results depends on the three key causal identification assumptions mentioned in the data analysis section. Although the FE regression eliminates all bias from time-constant omitted variables, the model remains vulnerable to omitted variable bias from left out time-varying variables. We have tried to minimize the risk of this by including a wide range of stressors across life domains, but important stressors may be missing. A risk of reverse causality and lagged effects of stressors on the outcome also exists, but because there are four years between the two waves, we do not consider this to be a serious source of bias, assuming that delayed effects of previous stressors and stress levels on current stressors and stress levels have tapered off during this period.

Another potential weakness of this study is the way we asked about stressors: "In the past 12 months, have you felt burdened by any of the following things?" We have no knowledge about

the actual or objective stressors, only the respondents' subjective appraisal of the burden represented by these stressors.

A final potential weakness is the selection bias introduced by response, dropout and item non-response. Those not included in the analytical sample differed on a number of sociodemographic characteristics and also had a higher mean score on PSS. However, selection bias was countered using a comprehensive weighting procedure.

## Conclusions

In this study, changes in perceived work- and non-work-related stressors along with changes in perceived social support caused changes in perceived stress. Changes in disease, work situation and social support had the greatest causal effect on changes in the perceived stress level among a wide range of stressors. Other stress factors also had a significant effect on the perceived stress level, whether individually or in combination with others. This points to the need for comprehensive policies to promote mental health that span life domains and include both the individual and the group as well as organizational and societal levels. Perceived work-related and non-work-related stressors along with perceived social support identified as drivers of perceived stress in this study may serve as specific targets in a comprehensive effort that aims at reducing stress both at individual and population level.

## Author Contributions

**Conceptualization:** Finn Breinholt Larsen, Jes Bak Sørensen.

**Data curation:** Finn Breinholt Larsen, Jes Bak Sørensen.

**Formal analysis:** Finn Breinholt Larsen.

**Funding acquisition:** Finn Breinholt Larsen.

**Investigation:** Finn Breinholt Larsen, Jes Bak Sørensen.

**Methodology:** Finn Breinholt Larsen.

**Project administration:** Finn Breinholt Larsen, Jes Bak Sørensen.

**Resources:** Finn Breinholt Larsen.

**Software:** Finn Breinholt Larsen.

**Writing – original draft:** Finn Breinholt Larsen, Jes Bak Sørensen.

**Writing – review & editing:** Finn Breinholt Larsen, Mathias Lasgaard, Morten Vejs Willert, Jes Bak Sørensen.

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
