## [Decision Letter · Decision Letter 0]

12 May 2023

PONE-D-23-05977Estimating the causal effects of work-related and non-work-related stressors on perceived stress level: a fixed effects approach using population-based panel dataPLOS ONE

Dear Dr. Larsen,

Thank you for submitting your manuscript to PLOS ONE. After careful consideration, we feel that it has merit but does not fully meet PLOS ONE’s publication criteria as it currently stands. Therefore, we invite you to submit a revised version of the manuscript that addresses the points raised during the review process.

We look forward to receiving your revised manuscript.

Kind regards,

Xiaozhao Yousef Yang, Ph.D.

Academic Editor

PLOS ONE

Journal Requirements:

Reviewers' comments:

Reviewer's Responses to Questions

**Comments to the Author**

1. Is the manuscript technically sound, and do the data support the conclusions?

Reviewer #1: Partly

Reviewer #2: Partly

2. Has the statistical analysis been performed appropriately and rigorously? 

Reviewer #1: Yes

Reviewer #2: Yes

3. Have the authors made all data underlying the findings in their manuscript fully available?

Reviewer #1: Yes

Reviewer #2: Yes

4. Is the manuscript presented in an intelligible fashion and written in standard English?

Reviewer #1: Yes

Reviewer #2: Yes

5. Review Comments to the Author

Reviewer #1: The manuscript is essentially a public health-style paper attempting to use panel data and fixed-effects models to identify stressors and to have practical implications for relieving psychological stress in contemporary society. The data is interesting, and the author provides a detailed report on the handling of missing values.

However, from a social science perspective, the manuscript has the following issues:

1.The article's theoretical dialogue is not clear, and there are some contradictions in the theoretical review, leading to a fuzzy storyline. The author tries to find two theoretical targets (Transaction Stress Model and Stress Process Model, TSM and SPM) in the introduction, only briefly explaining that neither can find the source of stress in theory before directly providing work-related and non-work-related stressors as the sources of stress in the article. However, as the author's states, SPM emphasizes social roles, which are themselves a source of stress. Violating role norms can lead to criticism and consequently, stress.

2.The article proposes two important concepts (spillover and crossover) to illustrate how stressors may cross various domains of life, but they are not evident in the variable measurement. The independent variables are uncorrelated with each other and do not cross domains of life.

3.The manuscript details how fixed-effects models can eliminate the confounding effect of non-time-varying omitted variables. However, statistical models exist to meet demand, and the use of fixed-effects models should specify explicit confounders, which the article does not do. Therefore, using fixed effects seems more like a mechanical program's job.

In summary, while the manuscript has some interesting data and analysis, it lacks clarity in its theoretical dialogue and the use of statistical models. The author should provide a more thorough and coherent theoretical foundation and clarify the use of fixed-effects models to ensure that they are not merely a mechanical program's work. The author should make a revolutionary revision.

Reviewer #2: Overall, the study is well-written with clear and concise structure. By collecting panel data and employing fixed effects regression, the authors investigated the impact of work-related and non-work-related stressors as well as perceived social support on perceived stress among workers. The study found that personal illness, working conditions, and lack of social support were the most significant stressors for workers, along with other relevant stress factors. Finally, the study proposed potential strategies that may help to manage stress and improve mental health among workers.

The study's main strengths lie in its valuable research topic and high academic and practical significance in understanding the impact of work and non-work-related stressors on workers' perceived stress. Furthermore, the use of panel data and fixed effects models effectively eliminates the influence of time-invariant individual traits, enhancing the credibility of the regression results and more accurately identifying the direct relationship between stressors and perceived stress.

However, the study's weakness is the inadequate explanation of the causal mechanism. As stated in the study, the interaction among different stressors is part of the research question. However, the analysis does not provide a clear presentation of the interaction between different stressors. Therefore, the final causal relationship appears somewhat weak and vague. The authors could consider further elaborating on this issue or improving their analytical framework to avoid confusion.

In conclusion, the study makes a valuable contribution to the field of stress management in the workplace. With the recommended improvements, the study can further enhance the understanding of the impact of stressors on workers' mental health and provide practical guidance for stress management.

6. PLOS authors have the option to publish the peer review history of their article (what does this mean?). If published, this will include your full peer review and any attached files.

Reviewer #1: No

Reviewer #2: No

---

## [Author Response · Author response to Decision Letter 0]

4 Jul 2023

Response to reviewers

Reviewer #1

The manuscript is essentially a public health-style paper attempting to use panel data and fixed-effects models to identify stressors and to have practical implications for relieving psychological stress in contemporary society. The data is interesting, and the author provides a detailed report on the handling of missing values.

We would like to express our gratitude for your review and the recognition you have given to our manuscript. Your feedback is truly appreciated.

However, from a social science perspective, the manuscript has the following issues:

1. The article's theoretical dialogue is not clear, and there are some contradictions in the theoretical review, leading to a fuzzy storyline. The author tries to find two theoretical targets (Transaction Stress Model and Stress Process Model, TSM and SPM) in the introduction, only briefly explaining that neither can find the source of stress in theory before directly providing work-related and non-work-related stressors as the sources of stress in the article. However, as the author's states, SPM emphasizes social roles, which are themselves a source of stress. Violating role norms can lead to criticism and consequently, stress.

Thank you for bringing this to our attention. Your observation regarding the "fuzzy" storyline is appreciated. Based on your comments, we have thoroughly revised the introduction section of the manuscript. 

We have replaced the following section (page 3) with the text below:

Research on stressors has, since its inception, specialized in many directions, thereby deepening our understanding of potential causes of stress [6]. Central questions are the following: What are the main sources of stress in contemporary societies, and how do stressors interact within and across areas of life? Previous studies suggest addressing stressors simultaneously in order to capture their additive or multiplicative effects [2, 6, 7]. In particular, studies are needed that include both work-related and non-work-related stressors [7-12].

New text:

Research on stressors has, since its inception, specialized in many directions, thereby deepening our understanding of potential causes of stress [6]. But the question still remains: What are the main sources of stress in contemporary societies? To answer this, it is necessary to study multiple stressors simultaneously as pointed out in previous studies [2, 6, 7]. In particular, studies are needed that include both work-related and non-work-related stressors [7-12]. Work-related stressors have been intensively researched, but they have predominantly been studied in isolation from non-work-related stressors. (Beauregard 2011, Marchand 2015). Similarly, studies of non-work stressors such as disease rarely include work-related stressors.

Furthermore, we have made improvements to the following sentence (page 4): 

In a previous study, we used a comprehensive approach and addressed a variety of stressor domains examining the relative importance of work-related and non-work-related stressors and perceived social support on overall perceived stress [7]. 

2. The article proposes two important concepts (spillover and crossover) to illustrate how stressors may cross various domains of life, but they are not evident in the variable measurement. The independent variables are uncorrelated with each other and do not cross domains of life.

Thank you for pointing out this ambiguity. The current study was not intended as an interaction study (we are in the process of preparing a study on how stressors cluster in different segments of the population). To clarify this, we have replaced the following sentence (page 5) with the sentence below:

The simultaneous or sequential interaction between stressors along with domain centrality, stress spillover and stress crossover emphasize the importance of an approach where a variety of stressors are examined jointly.

New sentence:

Although studies of domain centrality, stress spillover, and stress crossover do not per se aim to uncover the main sources of stress in modern society, by definition they include stressors from two or more societal contexts and have thereby expanded our understanding of the stress process compared to single-factor studies.

3. The manuscript details how fixed-effects models can eliminate the confounding effect of non-time-varying omitted variables. However, statistical models exist to meet demand, and the use of fixed-effects models should specify explicit confounders, which the article does not do. Therefore, using fixed effects seems more like a mechanical program's job.

In our view, the main strength of fixed-effects models is that the models control for unobserved time-constant factors. As pointed out by the reviewer, this does not exempt the authors from including relevant time-varying confounders in the models. We have considered the question thoroughly, but do not find variables in our dataset that we consider relevant to include in the model. It should be mentioned that our questionnaire regarding perceived stressors contains a catch-all question - "other stressors" - which should cover all unspecified stressors.

In summary, while the manuscript has some interesting data and analysis, it lacks clarity in its theoretical dialogue and the use of statistical models. The author should provide a more thorough and coherent theoretical foundation and clarify the use of fixed-effects models to ensure that they are not merely a mechanical program's work. The author should make a revolutionary revision.

Again, thank you for pointing this out. We hope that we have succeeded in making this clearer in the revised edition. We are a bit confused about the term 'revolutionary revision'.

Reviewer #2

Overall, the study is well-written with clear and concise structure. By collecting panel data and employing fixed effects regression, the authors investigated the impact of work-related and non-work-related stressors as well as perceived social support on perceived stress among workers. The study found that personal illness, working conditions, and lack of social support were the most significant stressors for workers, along with other relevant stress factors. Finally, the study proposed potential strategies that may help to manage stress and improve mental health among workers.

We sincerely appreciate your thorough review of our manuscript, and your positive assessment of the study.

The study's main strengths lie in its valuable research topic and high academic and practical significance in understanding the impact of work and non-work-related stressors on workers' perceived stress. Furthermore, the use of panel data and fixed effects models effectively eliminates the influence of time-invariant individual traits, enhancing the credibility of the regression results and more accurately identifying the direct relationship between stressors and perceived stress.

Thank you. Your recognition of the main strengths of our study is both encouraging and affirming.

However, the study's weakness is the inadequate explanation of the causal mechanism. As stated in the study, the interaction among different stressors is part of the research question. However, the analysis does not provide a clear presentation of the interaction between different stressors. Therefore, the final causal relationship appears somewhat weak and vague. The authors could consider further elaborating on this issue or improving their analytical framework to avoid confusion.

Thank you for pointing out this ambiguity in the wording of the abstract and introduction that could lead readers to expect that analyzes of interactions are part of the study. The study of interactions between stressors is not - and was not intended - as part of the aim of the study. We are planning a separate study on the clustering of stressors and the relationship between stress clusters and stress level (based on a latent class model). We have changed the wording of the abstract and introduction to avoid confusion.

We have replaced the following section (page 3) with the text below:

Research on stressors has, since its inception, specialized in many directions, thereby deepening our understanding of potential causes of stress [6]. Central questions are the following: What are the main sources of stress in contemporary societies, and how do stressors interact within and across areas of life? Previous studies suggest addressing stressors simultaneously in order to capture their additive or multiplicative effects [2, 6, 7]. In particular, studies are needed that include both work-related and non-work-related stressors [7-12].

New text:

Research on stressors has, since its inception, specialized in many directions, thereby deepening our understanding of potential causes of stress [6]. But the question still remains: What are the main sources of stress in contemporary societies? To answer this, it is necessary to study multiple stressors simultaneously as pointed out in previous studies [2, 6, 7]. In particular, studies are needed that include both work-related and non-work-related stressors [7-12]. Work-related stressors have been intensively researched, but they have predominantly been studied in isolation from non-work-related stressors. (Beauregard 2011, Marchand 2015). Similarly, studies of non-work stressors such as disease rarely include work-related stressors.

Furthermore, we have made improvements to the following sentence (page 4): 

In a previous study, we used a comprehensive approach and addressed a variety of stressor domains examining the relative importance of work-related and non-work-related stressors and perceived social support on overall perceived stress [7]. 

In conclusion, the study makes a valuable contribution to the field of stress management in the workplace. With the recommended improvements, the study can further enhance the understanding of the impact of stressors on workers' mental health and provide practical guidance for stress management.

We are pleased that you find our study to be a valuable contribution to the field of stress management in the workplace. Your recognition of the significance of our research is truly appreciated. We are particularly pleased that you highlighted the practical guidance for stress management that our study provides.

---

## [Editor Report · Decision Letter 1]

8 Aug 2023

Estimating the causal effects of work-related and non-work-related stressors on perceived stress level: a fixed effects approach using population-based panel data

PONE-D-23-05977R1

Dear Dr. Larsen,

We’re pleased to inform you that your manuscript has been judged scientifically suitable for publication and will be formally accepted for publication once it meets all outstanding technical requirements.

Kind regards,

Xiaozhao Yousef Yang, Ph.D.

Academic Editor

PLOS ONE
---

## [Editor Report · Acceptance letter]

17 Aug 2023

PONE-D-23-05977R1 

Estimating the causal effects of work-related and non-work-related stressors on perceived stress level: a fixed effects approach using population-based panel data 

Dear Dr. Larsen:

I'm pleased to inform you that your manuscript has been deemed suitable for publication in PLOS ONE. Congratulations! Your manuscript is now with our production department. 

Kind regards, 

on behalf of

Dr. Xiaozhao Yousef Yang 

Academic Editor

PLOS ONE